# Large room temperature spin-to-charge conversion signals in a few-layer graphene/Pt lateral heterostructure

Wenjing Yan [1], Edurne Sagasta[1], Mário Ribeiro[1], Yasuhiro Niimi[2], Luis E. Hueso [1,3] & Fèlix Casanova [1,3]

Electrical generation and detection of pure spin currents without the need of magnetic materials are key elements for the realization of full electrically controlled spintronic devices. In this framework, achieving a large spin-to-charge conversion signal is crucial, as considerable outputs are needed for plausible applications. Unfortunately, the values obtained so far have been rather low. Here we exploit the spin Hall effect by using Pt, a non-magnetic metal with strong spin-orbit coupling, to generate and detect pure spin currents in a few-layer graphene channel. Furthermore, the outstanding properties of graphene, with long-distance spin transport and higher electrical resistivity than metals, allow us to achieve in our graphene/Pt lateral heterostructures the largest spin-to-charge output voltage at room temperature reported so far in the literature. Our approach opens up exciting opportunities towards the implementation of spin-orbit-based logic circuits and all electrical control of spin information without magnetic field.

[1] CIC nanoGUNE, 20018 Donostia-San Sebastian, Basque Country, Spain. [2] Department of Physics, Graduate School of Science, Osaka University, 1-1 Machikaneyama, Toyonaka, Osaka 560-0043, Japan. [3] IKERBASQUE, Basque Foundation for Science, 48013 Bilbao, Basque Country, Spain. Wenjing Yan and Edurne Sagasta contributed equally to this work. Correspondence and requests for materials should be addressed to L.E.H. (email: l.hueso@nanogune.eu) or to F.C. (email: f.casanova@nanogune.eu)

A spintronic device with complete electrical functionality is attractive for its incorporation into the current charge-based integrated circuits. While advances have been made in the electrical control of spin transport[1–4], new approaches that allow electrical generation or detection of pure spin currents without using ferromagnetic materials (FM) as the spin source are also being developed[5–8]. In particular, in the emerging field of spin orbitronics[9, 10], by uniquely exploiting the spin-orbit coupling (SOC) in non-magnetic materials, spin-to-charge current conversions have been realized using the spin Hall effect (SHE)[11–13], the Rashba–Edelstein effect[14, 15], or the spin-momentum locking in topological insulators[16, 17]. Magnetization switching of FM elements for memories[5–7] or the recent proposal of a scalable charge-mediated non-volatile spintronic logic[18] are applications based on SOC, which can have a strong technological impact.

In the SHE, a pure spin current is generated from a charge current owing to the strong SOC in a non-magnetic material. Reciprocally, the inverse spin Hall effect (ISHE) can be used for spin detection, as a spin current is turned into a measureable charge current[19]. The efficiency of this spin-to-charge inter-conversion is given by the spin Hall angle ($\theta_{SH}$). As technological applications require large conversions, finding routes to maximize $\theta_{SH}$ has become a demanding task. The materials with highest reported yields are heavy metals with strong SOC, such as Pt[20, 21], Ta[5] and W[22, 23]. Recently, a clear route to enhance $\theta_{SH}$ of Pt has been unveiled[21]. Another path that has been explored to maximize the output voltage is the use of higher resistive spin Hall materials, although the potential enhancement is counteracted by the increased shunting of the induced charge current in the less resistive spin transport material, usually Cu or Ag[8]. It would be desirable to employ these spin-to-charge conversions to efficiently inject and detect pure spin currents in good spin transport materials with higher charge resistance.

Graphene, due to its weak SOC[24], is theoretically predicted and experimentally demonstrated to be an excellent material for spin transport[25–40]. Spin diffusion lengths of up to 90 μm have been reported at room temperature[31, 32], a parameter found to be fairly insensitive to temperature[37–39]. However, electrical spin injection from FM sources suffers from a lack of reproducibility owing to the required interfacial barriers[25, 35, 40], which are usually made of oxides that grow in an island mode on the graphene surface. An efficient and reliable spin injection and detection into such an outstanding spin transport material at room temperature using non-magnetic electrodes is of both fundamental and technological interests.

In our work, we demonstrate the generation (detection) of a pure spin current in a graphene-based lateral heterostructure by employing the SHE (ISHE) of Pt, avoiding the use of a FM source. Moreover, the large charge resistance of graphene as compared to the standard spin transport metals such as Cu and Ag eliminates completely the shunting effect, generating large output voltages. The spin-to-charge conversion signal in a graphene/Pt lateral device at room temperature is two orders of magnitude larger than the best performing ones previously reported that use metallic channels. Our concept of using charge-to-spin conversion to inject a spin current and spin-to-charge conversion to detect a spin currents in graphene-based devices could open future applications of all electrical control of spin information without magnetic field.

## Results

**Device structure.** We used the spin absorption method in lateral spin valve (LSV) devices to demonstrate spin generation and detection in graphene via the SHE and ISHE of Pt, respectively (see sketches in Fig. 1a). A scanning electron microscopy (SEM)

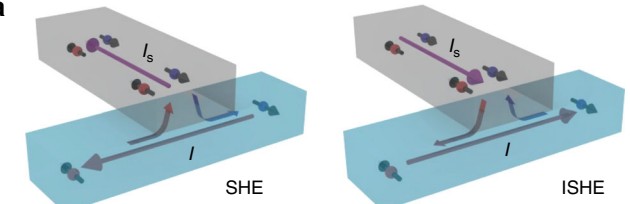

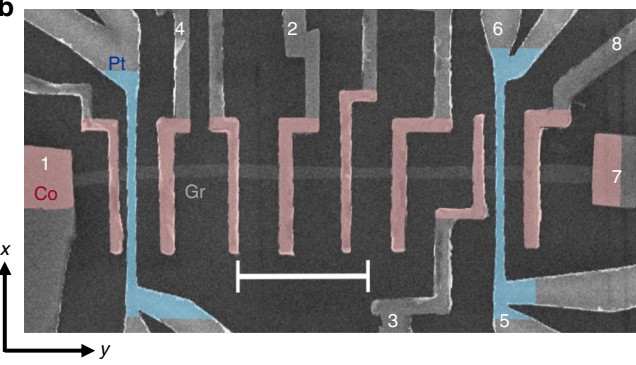

**Fig. 1** Illustration of the spin absorption method and scanning electron microscope (SEM) image of the device. **a** Sketches of spin injection (*left*) and detection (*right*) using the SHE and ISHE, respectively, with the spin absorption technique, in which a pure spin current $I_S$ is vertically transferred between a non-magnetic spin transport channel (*grey*) and a metal with strong SOC (*blue*). **b** SEM image of a graphene-based spintronic device. It consists of standard LSVs with ferromagnetic Co electrodes with TiO$_2$ barrier (*red color*) placed adjacent to each other. This electrode configuration allows the study of the spin transport properties of graphene (*grey color*) using standard FM electrodes (see Fig. 2a for details). In addition, Pt wires (*blue color*) are placed in between two pairs of Co electrodes. This extra configuration allows the study of spin absorption by Pt (see Fig. 3a for details) and spin current injection (via SHE) and detection (via ISHE) using Pt (see Fig. 4a for details). *Scale bar* is 3 μm

image of a complete device is shown in Fig. 1b. It consists of a 250-nm-wide flake of trilayer graphene (with a sheet resistance $R_{Gr}^{\blacksquare} = 1085\ \Omega$ and a carrier density $n \sim 8 \times 10^{11}\ cm^{-2}$) obtained via exfoliation[41], where spins are to be transported. Several ferromagnetic Co electrodes with their respective TiO$_2$ interfacial barriers are placed on top of the flake. The presence of TiO$_2$ between the Co electrode and the graphene channel leads to interface resistances between 10 and 42 kΩ. In addition, Pt wires are placed on top of the graphene channel. We use very resistive Pt with $\rho_{Pt} = 99\ (134)\ \mu\Omega cm$ at 50 (300) K and, therefore, a large spin Hall angle of $\theta_{SH} = 17.8 \pm 2.0\ (23.4 \pm 2.5)\%$[21]. Transport measurements are performed in a liquid-He cryostat with a superconducting magnet using a DC reversal technique[42–44]. See Methods for details on the device fabrication and measurements.

**Spin transport in a reference graphene LSV.** We first study the spin transport in a standard graphene LSV as shown in Fig. 2a. A spin-polarized current ($I_C$) is injected from a Co electrode into the graphene channel, creating a spin accumulation at the Co/graphene interface. This spin accumulation diffuses toward both sides of the graphene channel, creating a pure spin current, which is detected by another Co electrode as a nonlocal voltage ($V_{NL}$), see Fig. 2a. The non local resistance $R_{NL} = V_{NL}/I_C$ is high ($R_{NL}^P$, parallel) and low ($R_{NL}^{AP}$, antiparallel) depending on the relative orientation of the magnetization of the two electrodes, which can be set by applying an in-plane magnetic field in the $x$ direction ($B_x$) owing to the shape anisotropy of the electrodes (Fig. 2b). The difference $\Delta R_{NL} = R_{NL}^P - R_{NL}^{AP}$ is the spin signal. We obtain a spin

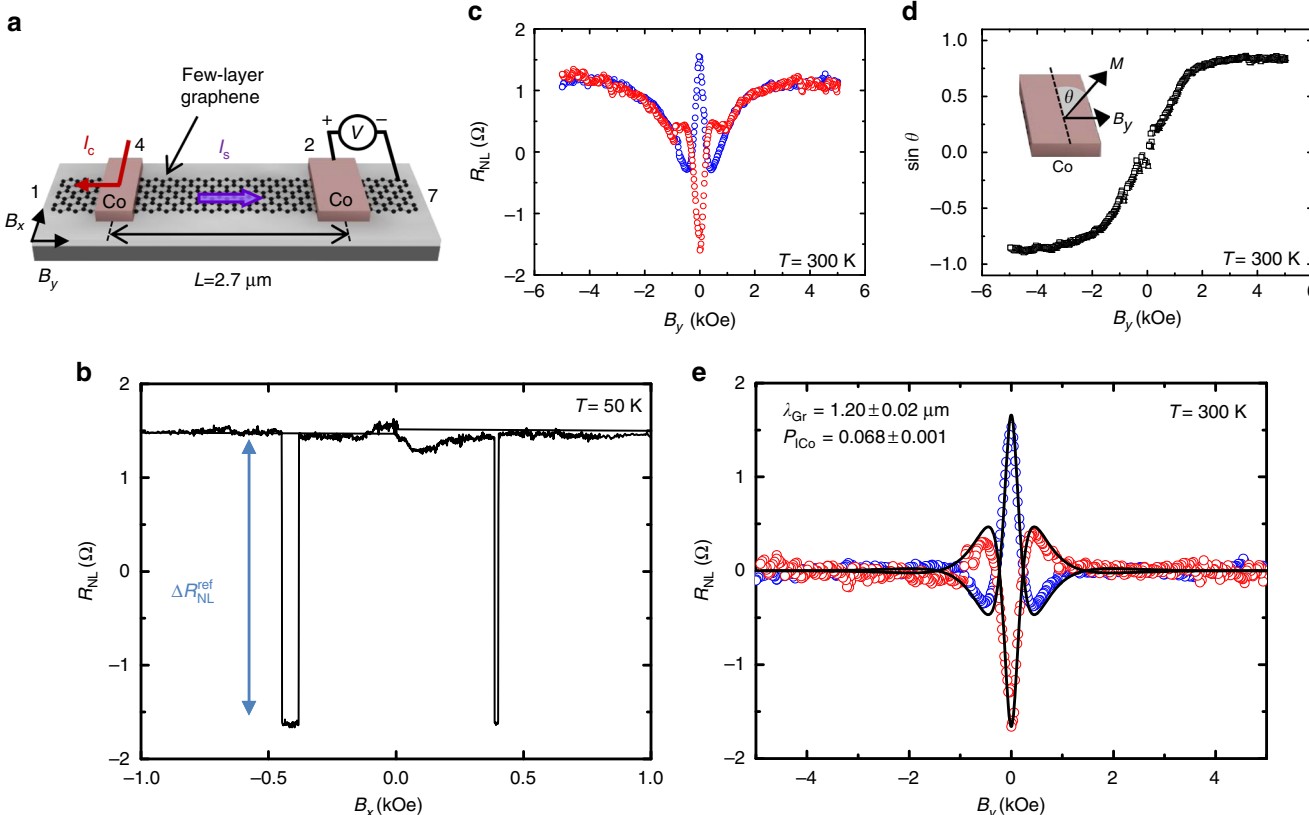

**Fig. 2** Spin transport in a reference trilayer graphene lateral spin valve. **a** Sketch of the measurement configuration, including the electrodes from Fig. 1b that are used, and the directions of the applied magnetic field ($B_x$ and $B_y$). **b** Non local resistance as a function of $B_x$ measured with $I_C = 10\ \mu\text{A}$ at 50 K and center-to-center Co electrode spacing $L = 2.7\ \mu\text{m}$ in the configuration shown in **a**. $R_{NL}$ switches between high and low resistance states for parallel and antiparallel magnetization orientation of the Co electrodes while sweeping $B_x$. The reference spin signal ($\Delta R_{NL}^{ref}$) is tagged. No baseline signal has been subtracted. **c** Hanle measurement, for which $R_{NL}$ is measured in the same device as a function of $B_y$ with $I_C = 10\ \mu\text{A}$ at 300 K in the configuration shown in **a** while the injecting and detecting Co electrodes are in the parallel (blue) and antiparallel (red) magnetization configurations. **d** $\sin \theta$ as a function of $B_y$ extracted from data in **c**. Inset: the magnetization direction of the Co electrode relative to x direction defines the angle $\theta$. **e** Pure spin precession and decoherence data extracted from data in **c**, where the contribution from the in-plane magnetization rotation of the electrodes under $B_y$ is removed. Spin transport properties are extracted by fitting the Hanle equation to the experimental data (black solid lines, see Supplementary Note 1)

signal of $\Delta R_{NL}^{ref} \sim 3\ \Omega$ owing to the large interface resistance given by good quality TiO$_2$. A Hanle measurement has been performed to characterize the spin transport properties of the graphene channel (Fig. 2c). As the injected spins are oriented along the x direction, a perpendicular in-plane magnetic field $B_y$ is applied. The precession and decoherence of the spins cause the oscillation and decay of the signal. In addition, the effect of the rotation of the Co magnetizations with $B_y$ tends to align the polarization of the injected spin current with the applied field, restoring the $R_{NL}$ signal to its zero-field value when the Co electrodes reach parallel magnetizations along the y direction at high enough $B_y$. By the proper combination of the measured $R_{NL}$ curves with an initial parallel (blue circles in Fig. 2c) and antiparallel (red circles in Fig. 2c) magnetization configuration of the electrodes in the x direction (see Supplementary Note 1), we can obtain the rotation angle $\theta$ of the Co magnetization (Fig. 2d) and the pure spin precession and decoherence (Fig. 2e). The data in Fig. 2e can be fitted using the Hanle equation[44, 45] (see Supplementary Note 1). The fitting allows us to extract the spin polarization of the Co/graphene interface $P_{ICo} = 0.068 \pm 0.001$ and the spin diffusion length of graphene $\lambda_{Gr} = 1.20 \pm 0.02\ \mu\text{m}$. Most importantly, the reference spin signals are independent of temperature (compare the amplitude of the signals in Fig. 2b at 50 K and Fig. 2e at 300 K), in agreement with the fact that $\lambda_{Gr}$ is basically insensitive to temperature[37–39]. In contrast, the spin diffusion length of metallic

channels such as Cu and Ag are significantly reduced with increasing temperature[42, 46].

**Spin absorption by Pt in a graphene LSV.** Once we have extracted the spin transport properties of graphene from a reference LSV, we now explore the spin absorption by Pt in the very same device. For this experiment, we use the non local configuration shown in Fig. 3a. A pure spin current in graphene is generated by spin injection from one Co electrode and detected by a second Co electrode, but in this case the pure spin current is partially absorbed by the Pt wire present in the middle of the spin current path before reaching the detector. The spin signal we measure after absorption by Pt is $\Delta R_{NL}^{abs} \sim 25\ \text{m}\Omega$, which is more than two orders of magnitude smaller than expected without the presence of the middle Pt wire (compare inset of Fig. 3b with Fig. 2b). This result indicates that the Pt wire acts as an extremely efficient spin absorber. We observe that $\Delta R_{NL}^{abs}$ has weak temperature dependence as it occurs in the reference LSV, implying that the Pt wire absorbs similar amount of spins across the temperature range investigated (see Fig. 3b).

**Spin generation and detection in a graphene/Pt lateral heterostructure.** After confirming that the Pt wire absorbs the spin current from graphene, and taking into account that Pt has a large $\theta_{SH}$[20, 21], our next experiment demonstrates we can indeed

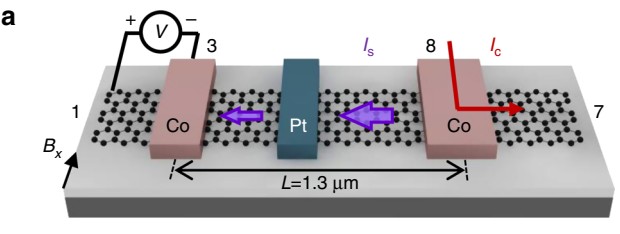

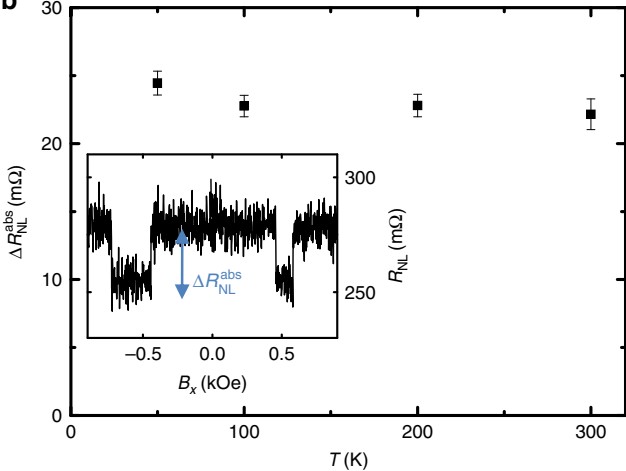

**Fig. 3** Spin absorption by Pt in a trilayer graphene lateral spin valve. **a** Sketch of the measurement configuration, including the electrodes from Fig. 1b that are used, and the direction of the applied magnetic field ($B_x$). **b** Spin signal after Pt absorption $\Delta R_{NL}^{abs}$ as a function of temperature. Inset: non local resistance as a function of $B_x$ measured with $I_C = 10 \, \mu A$ and center-to-center Co electrode spacing $L = 1.3 \, \mu m$ in the configuration shown in **a**, from which the values of $\Delta R_{NL}^{abs}$ are extracted for different temperatures. The curve shown corresponds to 50 K. Error bars are calculated using the standard errors associated with the statistical average of the nonlocal resistance in the parallel and antiparallel states

electrically detect this spin current by using the ISHE of the same Pt wire, employing the measurement configuration in Fig. 4a (*top sketch*). In this case, the pure spin current injected from the Co electrode diffuses along the graphene channel and is mostly absorbed by the Pt wire. In the Pt wire, owing to the ISHE, a charge current perpendicular to both the spin current direction and the spin polarization is created (Fig. 1a, *right*) and, thus, a voltage drop is generated along the Pt wire. The measured voltage normalized to the injected current $I_C$ yields the ISHE resistance, $R_{ISHE}$. By sweeping the magnetic field ($B_y$) from positive to negative, the magnetization of the Co electrode (as well as the orientation of the spin polarization) rotates, and $R_{ISHE}$ reverses sign as shown by the *blue curve* in Fig. 4b. According to the symmetry of the ISHE, the signal detected in the Pt wire should be proportional to $\sin \theta$[11, 13], a value which has been extracted from the Hanle data (Fig. 2d). Indeed, we observe a perfect match when overlapping the ISHE signal with $\sin \theta$ as a function of $B_y$ (Fig. 4b). This excellent match unambiguously confirms that the measured signal arises from spin-to-charge conversion. Other spurious effects such as magnetoresistance or heating are ruled out with control experiments (see Supplementary Note 2). The magnitude of the spin-to-charge conversion signal $\Delta R_{SCC}$ can be calculated by taking the difference between the saturation $R_{ISHE}$ resistance at large positive and negative field ($R_{ISHE}^+ - R_{ISHE}^- = \Delta R_{SCC}$).

The ISHE experiment shows that the Pt electrode can electrically detect spins travelling in the graphene channel. Next,

we demonstrate that a pure spin current can also be generated using the SHE of Pt and injected into graphene. Here, we pass a charge current $I_C$ through the Pt wire as shown in Fig. 4a (*bottom sketch*). The transverse spin current generated in Pt by the SHE has a spin polarization oriented along the $y$ axis, and the spin accumulation in the graphene/Pt interface leads to spin injection into graphene (Fig. 1a, *left*). By employing now the Co electrode as a detector, we are able to measure the pure spin current reaching the Co electrode as a voltage, obtaining the corresponding SHE resistance, $R_{SHE}$, after normalizing by $I_C$ (*black curve* in Fig. 4c). We observe that $R_{SHE}(B_y) = R_{ISHE}(-B_y)$ by swapping the voltage and current probes with the same polarity (see detailed electrode configurations in Fig. 4a), confirming the reciprocity between the ISHE and SHE in our experiment via the Onsager relation[13, 47]. The SHE and ISHE measurements demonstrate that it is possible to generate and detect pure spin currents in graphene using a non-magnetic spin Hall metal.

We have performed the ISHE experiment at different temperatures, as shown in Fig. 4d. Interestingly, as the temperature is increased from 10 to 300 K, $\Delta R_{SCC}$ increases from $\sim 5 \, m\Omega$ to $\sim 11 \, m\Omega$, indicating that the spin-to-charge conversion signal improves at higher temperatures. This increase of $\Delta R_{SCC}$ with temperature is robust and reproducible among different samples (Supplementary Note 3). Our devices based on the few-layer graphene/Pt heterostructure show superior performance over devices reported in literature using a metallic spin channel[8, 21, 48–52], as summarized in Fig. 4d. Two key aspects can be highlighted. In the first place, the $\Delta R_{SCC}$ signal measured in our devices is almost two orders of magnitude larger at 300 K. In the second place, the output signal in a graphene/Pt heterostructure increases significantly with increasing temperature in contrast to the decreasing tendency found when using a metallic channel.

## Discussion

Our experimental observations can be well explained by the standard one-dimensional spin diffusion model. The spin signal after absorption is given by the following equation (see Supplementary Note 4 for details):

$$\Delta R_{NL}^{abs} = 8 R_{Gr} Q_{ICo1} Q_{ICo2} P_{ICo}^2 \times$$

$$\frac{(Q_{IPt} + Q_{Pt}) e^{-\frac{L}{\lambda_{Gr}}}}{(2Q_{ICo1} + 1)(2Q_{ICo2} + 1) - 2(Q_{ICo1} + Q_{ICo2} + 1) e^{-\frac{L}{\lambda_{Gr}}} + e^{-\frac{2L}{\lambda_{Gr}}}},$$

$$(1)$$

where $Q_{Ik} = \frac{1}{1 - P_{Ik}^2} \frac{R_{Ik}}{R_{Gr}}$, being $R_{Ik}$ the resistance and $P_{Ik}$ the spin polarization of the $k$th metal/graphene interface. In our device, we define $k = Co1, Co2, Pt$ for the Co injector, Co detector and Pt wire, respectively, and we assume $P_{ICo1} = P_{ICo2} = P_{ICo}$. $R_{Gr} = \frac{R_{Gr}^\blacksquare \lambda_{Gr}}{w_{Gr}}$ is the spin resistance of graphene, where $R_{Gr}^\blacksquare$ is its sheet resistance. $Q_{Pt} = \frac{R_{Pt}}{R_{Gr}}$, being $R_{Pt} = \frac{\rho_{Pt} \lambda_{Pt}}{w_{Pt} w_{Gr} \tanh[t_{Pt}/\lambda_{Pt}]}$ the spin resistance of Pt. The geometrical factors $w_{Gr}$, $w_{Pt}$, $t_{Pt}$ and $L$ are the width of graphene, width of Pt, thickness of Pt and center-to-center distance between the Co electrodes, respectively. $\lambda_{Pt}$ is the spin diffusion length of Pt.

The spin-to-charge conversion signal $\Delta R_{SCC}$ of the ISHE experiment is given by the following expression:[21, 49, 50]

$$\Delta R_{SCC} = \frac{2 \theta_{SH} \rho_{Pt} x_{Pt/Gr}}{w_{Pt}} \left( \frac{\overline{I_S}}{I_C} \right). \qquad (2)$$

where $\overline{I_S}$ is the effective spin current injected vertically from graphene into the Pt wire that contributes to the ISHE in Pt and $x_{Pt/Gr}$ is the correction factor which considers the current in the

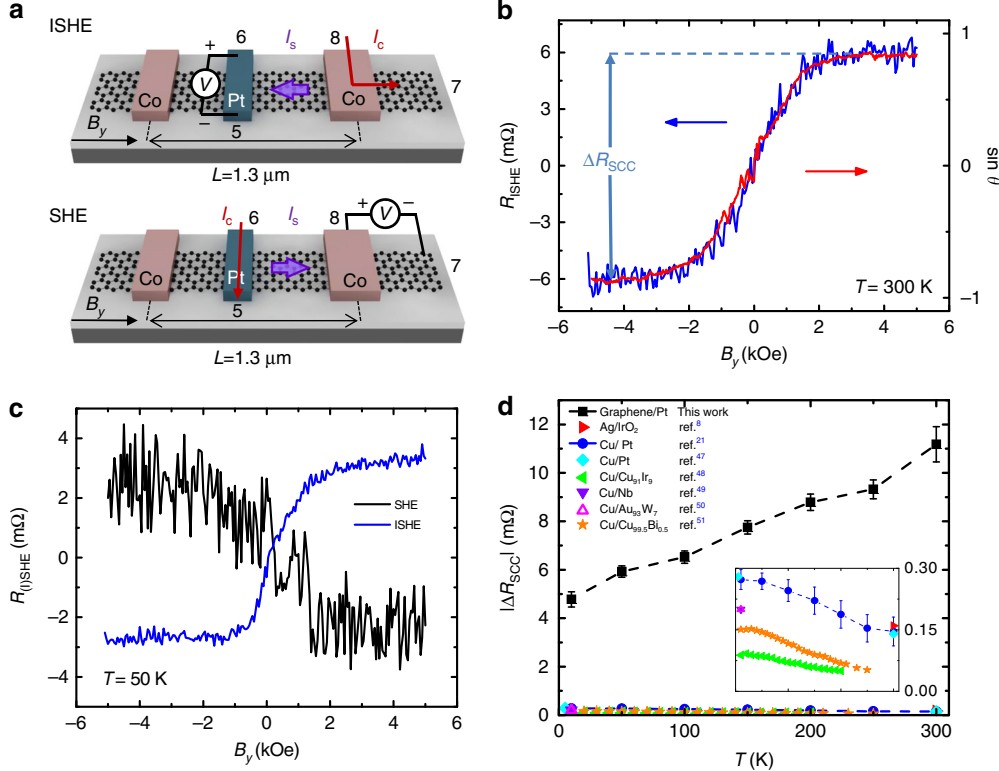

**Fig. 4** Spin-to-charge conversion in a trilayer graphene/Pt lateral heterostructure. **a** Sketch of the ISHE (*top*) and the SHE (*bottom*) measurement configurations, including the electrodes from Fig. 1b that are used, and the direction of the applied magnetic field ($B_y$). **b** ISHE resistance (*blue*) as a function of $B_y$ measured with $I_C = 10\,\mu A$ at 300 K. A baseline signal of 6.5 mΩ, corresponding to the Ohmic contribution given by the van der Pauw currents spreading into the voltage detector, has been subtracted. For comparison, sin θ (*red*) as a function of $B_y$ extracted from the Hanle measurement is also shown. The spin-to-charge conversion signal $\Delta R_{SCC}$ is tagged. **c** The ISHE (*blue*) and SHE (*black*) resistance as a function of $B_y$ measured with $I_C = 10\,\mu A$ at 50 K in the configurations sketched in **a** with center-to-center Co electrode spacing $L = 1.3\,\mu m$, showing the reciprocity of the two effects. A baseline signal of 4 mΩ (7 mΩ), corresponding to the Ohmic contribution, has been subtracted from the ISHE (SHE) curve. **d** Experimental values of $\Delta R_{SCC}$ at different temperatures measured in the graphene/Pt heterostructure. Literature values of $\Delta R_{SCC}$ of various spin Hall metals employing different metallic spin channels are also included for comparison: Ag/IrO$_2$[8], Cu/Pt[21, 48], Cu/Cu$_{91}$Ir$_9$[49], Cu/Nb[50], Cu/Au$_{93}$W$_7$[51] and Cu/Cu$_{99.5}$Bi$_{0.5}$[52]. Inset: Zoom of the main plot showing the data of the devices with metallic spin channels. *Error bars* are calculated using the standard errors associated with the statistical average of the nonlocal resistance at positive and negative saturated magnetic fields

Pt shunted through the graphene (see Supplementary Note 5 for details).

For the calculation, we substitute into eq. (1) and (2) the experimental values of $\Delta R_{NL}^{abs}$ and $\Delta R_{SCC}$, the obtained $P_{ICo}$ and $\lambda_{Gr}$ (Supplementary Note 1), the geometrical factors (measured from SEM images), and the values of $\rho_{Pt}$ and $\theta_{SH}$ of Pt[21]. We assume negligible current shunting into the graphene due to the much larger sheet resistance of graphene when compared to Pt at the junction area, $R_{Gr}^{\blacksquare} = 1085\,\Omega$ vs $\rho_{Pt}/t_{Pt} = 64\,\Omega$ at 300 K, which leads to $x_{Pt/Gr} \approx 1$. We extract two very sensitive parameters $\lambda_{Pt}$ and $R_{IPt}$, which are $2.1 \pm 0.4$ nm and $8.4 \pm 0.4\,\Omega$ at 300 K. The obtained $\lambda_{Pt}$ is expected when considering the resistivity of our Pt wire[21]. The small value of $R_{IPt}$ facilitates strong spin absorption by Pt from graphene and is compatible with our direct measurement (Supplementary Note 4). The good consistency of extracted values confirms that our assumption of $x_{Pt/Gr} \approx 1$ is robust.

Having quantified accurately all the parameters in our system, we can confirm the origin of the observed large spin-to-charge conversion and its strong temperature dependence. It mainly arises from four factors. First, the superior spin transport properties of graphene ($\lambda_{Gr} \sim 1.2\,\mu m$) and its temperature insensitivity. Graphene's exceptional ability to transport spins remains intact at room temperature, that is, the same amount of spin current arrives to the Pt absorber at different temperatures; Second,

although the amount of spin current to be converted remains the same, the efficiency of the conversion ($\theta_{SH}$) of Pt increases linearly with temperature from $17.8 \pm 2.0\%$ at 50 K to $23.4 \pm 2.5\%$ at 300 K[21]; Third, the resistivity of Pt increases from 99 µΩcm at 50 K to 134 µΩcm at 300 K; and fourth, the negligible shunting of the charge current in Pt by graphene ($x_{Pt/Gr} \approx 1$). The enhancement of $\Delta R_{SCC}$ with increasing temperature mainly benefits from the first three factors, which are constant $\lambda_{Gr}$ and increasing $\theta_{SH}\rho_{Pt}$ product as described in eq (2). In contrast, in metallic spin channels, the spin diffusion length of the metal channel decreases significantly with increasing temperature[42, 46], leading to reduced output voltage. Our devices give much larger $\Delta R_{SCC}$ than those using metallic spin channels mainly due to the first (long spin diffusion length of graphene) and fourth (negligible shunting) factors. In traditional metallic LSV devices, the resistivity of the metal channel is close or smaller than that of the spin Hall metal, thus $x$ are much lower (0.05–0.36)[21, 52], a serious issue preventing large spin-to-charge conversion pointed out recently[8]. However, in our device with few-layer graphene/Pt heterostructure, $x_{Pt/Gr} \approx 1$ is close to ideal and the use of more resistive graphene (single or bilayer) is not necessary, as $x_{Pt/Gr}$ cannot be further increased. Further improvement to the spin-to-charge conversion could be easily achieved by using high quality graphene devices, where almost two orders of magnitude enhancement of $\lambda_{Gr}$ is obtained[31, 32], or reducing the spin current dilution into the Pt

wire by decreasing its thickness (as can be deduced from Supplementary Equation 10).

Finally, it is worth mentioning that a direct comparison between $\Delta R_{NL}^{ref}$ ( ~ 3 Ω) and $\Delta R_{SCC}$ ( ~ 12 mΩ) is not appropriate, because they quantify different outputs. Whereas the former only probes the spin accumulation in the channel through a ferromagnetic tunnel barrier and acts as spin detector, the latter is a measurement of the converted charge current through a transparent interface, which can be potentially utilized (See Supplementary Note 6 for an extended discussion).

To conclude, we succeeded in electrically injecting and detecting pure spin currents in few-layer graphene by employing the SHE and ISHE of a non-magnetic material, respectively. The extraordinary ability of graphene to transport spins, together with its relatively high resistance compared to Pt, results in the largest spin-to-charge conversion signal reported so far. Most importantly, the largest conversion, which is two orders of magnitude larger than in devices employing metallic spin channels, occurs at room temperature. The fuse and perfect match of these two elements in a heterostructural device of graphene/Pt provides new plausible opportunities for future spin-orbit-based devices.

## Methods

**Device fabrication**. To fabricate our devices, few-layer graphene flakes are first produced by micromechanical cleavage of natural graphite onto 300-nm-thick $SiO_2$ on doped Si substrate using Nitto tape (Nitto SPV 224P) and identified using its optical contrast[41]. We select flakes with the most convenient shape (long and narrow), regardless of the number of layers, as the excellent spin transport properties do not depend strongly on the number of graphene layers[36]. The nanofabrication of the device follows two steps of e-beam lithography with electrode metal deposition and lift-off. For the 200-nm-wide Pt wires, 21 nm of Pt were sputtered at $0.6$ Ås$^{-1}$ using 40 W in 3 mTorr of Ar pressure. This deposition condition gives rise to very resistive Pt with $\rho_{Pt} = 99$ (134) μΩcm at 50 (300) K. The 35-nm-thick Co electrodes with widths between 150 and 350 nm are deposited in an ultra-high vacuum chamber using e-beam evaporation on top of 6 Å of Ti after the natural oxidation of Ti in air. The presence of $TiO_2$ between the Co electrode and the graphene channel leads to interface resistances between 10 and 42 kΩ.

**Electrical measurements**. The measurements are performed in a Physical Property Measurement System by Quantum Design, using a DC reversal technique with a Keithley 2182 nanovoltmeter and a 6221 current source[42–44].

**Data availability**. All relevant data are available from the authors.

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

## Acknowledgements

This work was supported by the European Union 7th Framework Programme under the Marie Curie Actions (607904-13-SPINOGRAPH), by the Spanish MINECO under Project No. MAT2015-65159-R, by the Basque Government under Project No. PC2015-1-01, and by the Regional Council of Gipuzkoa under project No. 100/16. E.S. thanks the Spanish MECD for a Ph.D. fellowship (Grant No. FPU14/03102).

## Author contributions

F.C. conceived the study. W.Y., E.S and M.R. performed the experiments. W.Y., E.S., L.E.H. and F.C. analysed the data, discussed the experiments and wrote the manuscript; Y.N. derived the equations used in the analysis of the spin absorption technique. All the authors contributed to the scientific discussion and manuscript revision. L.E.H. and F.C. co-supervised the work.

## Additional information

**Competing interests:** The authors declare no competing financial interests.

