## [Peer Review File · Nature Communications]

Reviewers' comments:

Reviewer #1 (Remarks to the Author):

The authors have reported their finding on the spin-to-charge conversion in few-layer graphene/Pt lateral hetero-structures. When using non-magnetic Pt wire on the top of graphene, they have demonstrated one hand the spin injection with SHE and the other hand the spin detection with ISHE in non-local configuration. The spin-to-charge conversion signal is found to be two orders larger than those in metallic system and the signal increases with the increasing of temperature, which favors future room temperature spin-orbit based spintronics applications. The paper is well structured and their results are interesting and original. However the present status of paper is immature for the publication in Nat. Comm. The authors should provide more supplementary results to convince their conclusions.

1. The thickness of graphene should be identified even though the authors claim that the spin transport property does not depend on the thickness. Another important issue not yet addressed is the contact property of Pt/graphene, especially with the employed sputtering growth technique. The contact resistance will dramatically modify the conductivity mismatch condition and change the spin-injection and spin-detection efficiency. Two terminal IV curves could be useful get the information.
2. The authors have used 21nm thick Pt wire for spin-injection and spin-detection. This thickness is the optimal condition or not? The authors should provide Pt thickness dependent results. Most of literatures show that the largest spin-signal often happens when the thickness of Pt is close to its spin diffusion length 6-10 nm.
3. The authors should provide the measurement on at least one control sample with non-magnetic and small spin-orbit coupling contact (eg. Al, Cu...).
4. Samples with different distance between Pt and Co contacts should be provided to show an exponential decay of SHE or ISHE signals, which could be used to check the measured 1.2um spin diffusion length in graphene.
5. The authors should explain why the amplitude of SHE signal should be the same as the amplitude of ISHE signal but opposite sign with the Onsager relationship. In the fig.4c and supplementary fig.1a, is the same current density used for the injection from Co contact to graphene and for SHE in the Pt wire? Why the ΔR_{SCC} in the second device is 4 times smaller than Device 1 in the main text? Which differences exist between them?
6. The increase of ΔR_{SCC} with temperature is more than 2 times from 50K to 300K. However as authors demonstrated, the spin relaxation is constant in graphene while the spin hall angle increases only 30% from 50K to 300K. Some other reasons should be responsible for this increase. Do the authors check the increase of resistivity of Pt wire with temperature?
7. A baseline signal has been subtracted from ISHE (SHE) curves. Can the authors explain where the background signals come from? When changing the sign of current in the Pt wire, this background will change the sign or not?
8. The SHE and ISHE curves should be presented with increasing field and also decreasing field to exclude background shift effect (due to heating).
9. The efficiency of spin-injection and spin-detection with SHE and ISHE in Pt wires seems quite low by comparing the ΔR_{SCC} ($\sim 12m\Omega$) and non-local ΔR_{NL} ($\sim 3\Omega$). A discussion should be added on how to enhance this efficiency in the future? Is it only due to the impedance mismatch problem?

Reviewer #2 (Remarks to the Author):

This paper describes a relevant experimental problem: how to convert spin currents into charge voltages with optimum efficiency. The authors do experiments where they inject spins into few-layer graphene with conventional ferromagnetic tunnel junctions, and they detect the spin currents via the inverse spin Hall effect in a Pt electrode. I notice the following:

1) They compare their systems with other systems, and conclude that their few-layer graphene system has the largest efficiency for conversion of spin currents into charge voltages. In my opinion that is correct. However their system is a very specific one, so it makes it unclear whether this "figure of merit" can really be compared between different systems.

2) The signals they obtain are a few milliohms, and they are small compared to what can be achieved with conventional detection/injection (using ferromagnetic tunnel barriers). Also they have shown that when they use Pt strips for both injection and detection, the signal is too small to be measured. Although this is a nice "proof of principle" demonstration, I therefore think that the practical applications will be limited.

3) On page 9 they write "The excellent match between the two curves unambiguously confirms that the signal arises from spin-to-charge conversion". I disagree. In the linear transport regime, where most of these experiments are done, reciprocity relations for four terminal measurements should always be obeyed, no matter what the mechanisms are. So the observation of reciprocity does not exclude other mechanisms, such as stray fields. They can be relevant because when the Co ferromagnet is magnetized in the y-direction, it will produce stray fields in the graphene close to the ends of the electrodes. These will have components in the z-direction, so they could produce magneto resistance effects which would scale with the magnetization in the y-direction of the Co. I did not see a discussion of this. Can the authors please argue why the effects of stray fields are not relevant?

4) According to the authors the spin-charge- conversion at the Pt strip happens because of spin which are polarized in the y-direction enter the strip in the z-direction, thus producing a charge current in the x-direction. However I would not expect this to happen uniformly over the entire Pt/graphene interface. Because this is a metallic interface I would expect that the spins will already enter at the beginning of the strip. It is not clear if the authors have taken this into account, or whether it changes their conclusions. Can the authors comment on this?

In summary, I am willing to give a positive recommendation, but I first would like to see a convincing reply to my points 3) and 4)

Reviewer #1 (Remarks to the Author):

The authors have reported their finding on the spin-to-charge conversion in few-layer graphene/Pt lateral hetero-structures. When using non-magnetic Pt wire on the top of graphene, they have demonstrated one hand the spin injection with SHE and the other hand the spin detection with ISHE in non-local configuration. The spin-to-charge conversion signal is found to be two orders larger than those in metallic system and the signal increases with the increasing of temperature, which favors future room temperature spin-orbit based spintronics applications. The paper is well structured and their results are interesting and original. However the present status of paper is immature for the publication in Nat. Comm. The authors should provide more supplementary results to convince their conclusions.

1. The thickness of graphene should be identified even though the authors claim that the spin transport property does not depends on the thickness.

The flake is trilayer. We also recognize the importance of this information for reproducibility of our work by other groups and hence the thickness was already identified clearly for the device reported in the original paper (Page 3, “Device structure” paragraph and in captions of Figs. 2, 3 and 4).

Another important issue not yet addressed is the contact property of Pt/graphene, especially with the employed sputtering growth technique. The contact resistance will dramatically modify the conductivity mismatch condition and change the spin-injection and spin-detection efficiency. Two terminal IV curves could be useful get the information.

We totally agree with the referee that the contact resistance is an extremely important issue, this is why it was already discussed and addressed in detail in the original manuscript. We took great care regarding this parameter, studying it both experimentally and by fitting and obtaining compatible results:

i) First of all, we measured the interface resistance using a direct 4 point measurement. The measured values are negative, which is an artifact due to the very small interface resistance (a few ohms) comparing to graphene ($\sim 1 \text{ k}\Omega$). This was widely discussed at the end of Supp. Note 3 (now Supp. Note 4). For this reason, a direct measurement done with two terminal IV curves as suggested by the referee will not be useful, because the resistance of the graphene channel is much larger ($\sim 1 \text{ k}\Omega$) than the one of the interface ($\sim 10 \Omega$).

ii) In order to evaluate the very small interface resistance, so small that we could not even measure, we fit our results of spin absorption and spin-to-charge conversion to the spin diffusion model (Eqs. 1 and 2) to obtain precisely this parameter, the contact resistance, which is $\sim 10 \Omega$. We recognized the importance of the parameter in pag. 12 (“We extract two very sensitive parameters λ_{Pt} and R_{IPt} [...]”) and in pag. 13 (“The small value of R_{IPt} facilitates strong spin absorption by Pt from graphene”).

2. The authors have used 21nm thick Pt wire for spin-injection and spin-detection. This thickness is the optimal condition or not? The authors should provide Pt thickness dependent results. Most of literatures show that the largest spin-signal often happens when the thickness of Pt is close to its spin diffusion length 6-10 nm.

This thickness is not in the optimal condition, and this was clearly stated in the original text (pag. 14: “further improvement to the spin-to-charge conversion could be easily achieved by [...] reducing the spin current dilution into the Pt wire by decreasing its thickness (as can be deduced from Supplementary Eq. 10)”).

The referee is right that the largest spin-to-charge conversion signal should happen when the thickness of Pt is close to its spin diffusion length. More precisely, this effect is essentially taken into account by the prefactor of Supplementary Eq. 10:

$$\frac{\lambda_M}{t_M} \frac{\left(1 - e^{-\frac{t_M}{\lambda_M}}\right)^2}{1 - e^{-\frac{2t_M}{\lambda_M}}}$$

which is plotted here:

The prefactor is in fact maximum (0.5) when $\frac{t_M}{\lambda_M} = 0$ and decreases with increasing $\frac{t_M}{\lambda_M}$. For our case, $\frac{t_M}{\lambda_M} = 10$ and the prefactor is 0.1. Therefore, the signal is “only” 5 times smaller than the maximum value. At a first look, it seems one could decrease the thickness of the Pt nanowire to improve the prefactor. If we assume, for the sake of simplicity, that we can have a 10-nm-thick Pt wire with the same resistivity, the prefactor would be 0.2, i.e., the spin-to-charge conversion signal doubles. In reality, it is not so straightforward because the resistivity of Pt increases when reducing the thickness, leading to a decrease of λ_{Pt} [see Refs. 20 and 21]. It is thus not a simple task to obtain a thickness to optimize this prefactor.

Carrying out Pt thickness dependent experiments is thus a tedious task that will not produce any inspiring ideas and, furthermore, will not increase the novelty of our work by any means as we are demonstrating a proof of principle device.

3. The authors should provide the measurement on at least one control sample with non-magnetic and small spin-orbit coupling contact (eg. Al, Cu...).

We performed the control experiment by replacing Pt with Cu in a device similar to that of Fig. 1b as suggested by the referee. We first measured the lateral spin valves in a nonlocal geometry to ensure that there is spin injection from the Co electrode next to the Cu wire to the graphene channel (New Supplementary Fig. 1a). We then measured the voltage drop across the two ends of the Cu wire in the ISHE measurement geometry, as we did for Pt in the main manuscript (New Supplementary Fig. 1b). As expected, all we can see is a flat line. The same results are obtained in 4 other control samples.

A new section regarding the control sample and discussion is added in a new Supplementary Note 2. For the convenience of the referee, the figures are reproduced here:

4. Samples with different distance between Pt and Co contacts should be provided to show an exponential decay of SHE or ISHE signals, which could be used to check the measured 1.2μm spin diffusion length in graphene.

The referee is right to think along this line. However, we are afraid that, for practical reasons, this experiment is not reliable. Lateral spin valves in graphene were first reported 10 years ago (Ref. 37) and, ever since then, the only way to reliably extract the spin diffusion length of the graphene has been with the Hanle effect as we are doing here (spin signal as a function of a perpendicular magnetic field for a single distance between ferromagnetic (FM) electrodes). The reason is that, unlike lateral spin valves in other systems (metals, semiconductors), electrical spin injection from FM sources suffers from a lack of reproducibility due to the required interfacial barriers, which are usually made of oxides that grow in an island mode on the graphene surface. This lack of reproducibility occurs even on the same graphene flake and prevents the comparison between different distances even in the same device [see for instance Ref. 40 or the review paper F. Volmer et al., *Synthetic Metals* 210, 42 (2015)]. In our device, the interface resistance varies within a narrow range of 10-42 kΩ, which is stated in page 4. It is a long standing in the research community. Our measured Hanle effect (Fig. 2c) is thus the standard and so far only way to reliably extract the spin diffusion length of graphene.

In order to extract the spin diffusion length accurately from fitting of the Hanle effect, we consider the interface resistances between graphene and Co, otherwise it would lead to an underestimation of the spin diffusion length [T. Maassen et al., *PRB* 86, 235408 (2012) and Supp. Note 1].

In any case, the spin diffusion length we obtain in our device is a typical value reported in the literature for graphene on SiO₂, see the review article by W. Han *et al.* *Nature Nanotechnology* 9, 794 (2014).

5. The authors should explain why the amplitude of SHE signal should be the same as the amplitude of ISHE signal but opposite sign with the Onsager relationship.

Onsager's reciprocity relations relate linear response coefficients between flux densities and thermodynamic forces to one another. They are based on the fundamental principle of microreversibility for systems with time-reversal symmetry (TRS). When TRS is broken, microreversibility further requires all TRS breaking fields to be inverted, such as magnetic fields (B). In this case, the Onsager reciprocity relations read $L_{ij}(B) = L_{ji}(-B)$ where the linear coefficient L_{ij} determines the response of the flux density J_i , for instance the spin current, to a weak thermodynamic force X_j , for instance the electric field. For the case of SHE and ISHE, and for our configuration, the linear coefficient is the non-local resistance and therefore $R_{SHE}(B_y) = R_{ISHE}(-B_y)$.

This explanation has been inserted as a new reference (Ref. 52) in the main text.

In the fig.4c and supplementary fig.1a, is the same current density used for the injection from Co contact to graphene and for SHE in the Pt wire?

We did not explicitly mention the dimensions of Device #2, but they are the same as Device #1. Width of Co is 344 nm in Device #1 (336 nm in Device #2) and widths of Pt is 198 nm (193 nm in Device #2). Since the current is the same (10 μ A) in the SHE and ISHE experiments, as stated in both figure captions, the current densities are not the same. However, the current density is irrelevant in this case, because the non-local resistance is a measured voltage normalized to the injected current and, thus, it should not depend on the magnitude of the current or current density applied. Due to reciprocity as discussed above, ISHE and SHE gives the same spin-to-charge conversion signal.

We have added the dimensions of Device #2 in the new Supplementary Note 3.

Why the ΔR_{SCC} in the second device is 4 times smaller than Device 1 in the main text? Which differences exist between them?

As we stated clearly in the old Supp. Note 2 (now Supp. Note 3): “The magnitude of the ΔR_{SCC} signal measured in Device #2 is smaller than the device presented in the main text (#1) (compare Supplementary Fig. 1b (now 3b) with Fig. 4d). This is due to the variation of the interface resistances between Co and graphene, i.e., R_{ICo1} in Device #2 is smaller (2 k Ω) than in Device #1 (14.7 k Ω).” The smaller the interface resistance is, the less efficient the spin injection is from Co into graphene due to the conductivity mismatch problem. Therefore, in Device #2 there is less spin current to be converted into charge current in the Pt. It is precisely for this same reason that one could not extract spin diffusion length reliably from distance dependent measurement, which we discussed in point 4 of the referee’s comment.

6.The increase of ΔR_{SCC} with temperature is more than 2 times from 50K to 300K. However as authors demonstrated, the spin relaxation is constant in graphene while the spin hall angle increases only 30% from 50K to 300K. Some other reasons should be responsible for this increase. Do the authors check the increase of resistivity of Pt wire with temperature?

We agree with the referee: we oversimplified our statement in page 13. The value of the resistivity of Pt, which increases by 35% from 99 $\mu\Omega$ cm at 50 K to 134 $\mu\Omega$ cm at 300 K as we reported in page 4, also helps in the increase of ΔR_{SCC} with temperature. It is the overall product of the spin hall angle times the resistivity that is proportional to ΔR_{SCC} (see Eq. 2) and this product goes from 17.6 $\mu\Omega$ cm at 50 K to 31.3 $\mu\Omega$ cm at 300 K (an 80% increase, which accounts for most of the observed 90% increase of ΔR_{SCC}).

We have modified the text in the discussion section of the manuscript in order to include all the contributing factors giving rise to our large signal and its temperature dependence.

7.A baseline signal has been subtracted from ISHE (SHE) curves. Can the authors explain where the background signals come from? When changing the sign of current in the Pt wire, this background will change the sign or not?

This background is the classical van der Pauw current spreading from the injecting terminals. By injecting a current, a net current will reach the detection terminals, with magnitude decreasing exponentially with the distance to the injecting terminals. This Ohmic contribution in the device scheme presented is determined using the expression, $R_{NL,Ohmic} = \frac{\rho_{xx}}{\pi} \ln \left[\frac{\cosh(\pi L/W)+1}{\cosh(\pi L/W)-1} \right]$, which for cases where $L > W$ is usually approximated as

$R_{NL,Ohmic} \approx \frac{4}{\pi} \rho_{xx} \exp\left(-\pi \frac{L}{W}\right)$. Because the voltage measured is linear with the injected current (Ohmic), the non-local resistance does not change sign. Unlike the backgrounds arising from heat dissipation, this background is not suppressed when using the usual ac lock-in or dc reversal techniques [see, for instance, Casanova et al. PRB 79, 184415 (2009)].

Following the referee's suggestion, we have explained the origin of the background signals where they are mentioned in the text (caption of Fig. 4).

8. The SHE and ISHE curves should be presented with increasing field and also decreasing field to exclude background shift effect (due to heating)

We agree with the referee that this is an important check. Although we do not expect heating due to the small current (10 μ A) we are applying, we did perform these measurements as a control and make sure that the curves with increasing and decreasing field match well. We did not add those extra curves in the submitted manuscript for the sake of simplicity.

An example of these curves at 300 K measured in Device 1 is now added to new Supplementary Note 2. For the convenience of the referee, the figure is shown below:

9. The efficiency of spin-injection and spin-detection with SHE and ISHE in Pt wires seems quite low by comparing the ΔR_{SCC} (~ 12 m Ω) and non-local ΔR_{NL} (~ 3 Ω). A discussion should be added on how to enhance this efficiency in the future? Is it only due to the impedance mismatch problem?

First of all, one should be careful when directly comparing the non-local ΔR_{NL} signal and the spin-to-charge conversion ΔR_{SCC} signal. The former only probes the spin accumulation in the channel (in this case graphene) through a tunnel barrier or high resistive interface leading to a large voltage drop, but it cannot be further utilized, for instance to convert it to charge current for cascading in a spin-based logic circuit or to directly switch a magnetic element via spin transfer torque of the pure spin current. This limitation is equivalent to that observed in the local magnetoresistance of a spin valve: a high resistive interface helps in the spin injection, but is detrimental for the spin detection, because the current cannot flow into the detector [see Fig. 3 in Fert and Jaffrès, PRB 64, 184420 (2001)]. On the other hand, the configuration of the spin-to-charge conversion consists of a transparent interface through which spins can be absorbed or injected. Here the transport is diffusive and the referee is right that the impedance mismatch problem plays a role here. But the transparent interface is necessary to allow for absorption of the pure spin current in the spin Hall material, which is then converted to a charge current which can be potentially utilized. ΔR_{SCC} directly probes the charge current generated in the spin Hall metal.

After this clarification on why a direct comparison between ΔR_{SCC} ($\sim 12\text{m}\Omega$) and non-local ΔR_{NL} is not appropriate, it is true that ΔR_{SCC} can be further optimized. Nevertheless, a discussion on how to enhance the efficiency was already present in original page 13 (now page 14): “However, further improvement to the spin-to-charge conversion could be easily achieved by using high quality graphene devices, where almost two orders of magnitude enhancement of λ_{Gr} is obtained^{31,32}, or reducing the spin current dilution into the Pt wire by decreasing its thickness (as can be deduced from Supplementary Eq. 10)”.

In any case, the referee is completely right that the main limitation in enhancing the efficiency is the impedance mismatch, which is an unavoidable toll when the use of transparent interfaces is required for the envisioned application.

We have now added this discussion in the discussion section of the main text and in the new Supplementary Note 6.

Reviewer #2 (Remarks to the Author):

This paper describes a relevant experimental problem: how to convert spin currents into charge voltages with optimum efficiency. The authors do experiments where they inject spins into few-layer graphene with conventional ferromagnetic tunnel junctions, and they detect the spin currents via the inverse spin Hall effect in a Pt electrode. I notice the following:

1) They compare their systems with other systems, and conclude that their few-layer graphene system has the largest efficiency for conversion of spin currents into charge voltages. In my opinion that is correct. However their system is a very specific one, so it makes it unclear whether this “figure of merit” can really be compared between different systems.

We think ΔR_{SCC} is a good figure of merit and the comparison between different systems is valid for the following reasons:

- 1) In all systems that we compare, the same type of device is used to perform the same type of measurement, which is a non-local ISHE/SHE configuration in a lateral spin valve). In all of them, the spin current is transported using a non-magnetic channel to the spin Hall metal where it is absorbed and converted into charge current. ΔR_{SCC} gives the output that corresponds to this spin-to-charge conversion; where ΔR_{SCC} is the measured voltage drop along the spin Hall metal normalized with respect to the injected current. Therefore, ΔR_{SCC} is a good figure of merit to be compared among all these systems.
- 2) The difference is the combination of materials in different systems. However the material selection itself does not change the nature of the measurement or the physical quantity that we are monitoring. The only thing it does it to optimize the output signal ΔR_{SCC} by exploring the best performance of each constituent material, for example, spin Hall angle, resistivities, spin diffusion lengths, etc.. In the end, it is the engineering of nanodevices, which is precisely the added flexibility and novelty of 2D materials because it allows for atomically thin layers to be manipulated on the macroscopic scale.
- 3) Furthermore, for spin-to-charge conversion based technological applications, the largest possible ΔR_{SCC} is required. Therefore, a direct comparison of this magnitude among different kind of devices is appealing and convenient.

2) The signals they obtain are a few milliohms, and they are small compared to what can be achieved with conventional detection/injection (using ferromagnetic tunnel barriers). Also they have shown that when they use Pt strips for both injection and detection, the signal is too small to be measured. Although this is a nice “proof of principle” demonstration, I therefore think that the practical applications will be limited.

We refer here to the answer of point 9 by referee 1, where we explain why non-local signal and the spin-to-charge conversion signal cannot be directly compared.

We agree with the referee that a full spin injection and detection with Pt is not useful at this stage. Nevertheless, the combination of spin injection from one ferromagnetic element (where the non-volatile information is stored) and subsequent spin-to-charge current conversion in a non-magnetic element is important for cascading in potential applications such as spin-orbit logic [Ref. 18] or also to avoid dealing with the relative magnetic orientation of a second ferromagnet when used as a detector. And, in this case, we showed that the overall spin-to-charge conversion of our proof-of-principle device is 2 orders of magnitude more efficient than previous devices.

This discussion is now added in the discussion section of the main text and in the new Supplementary Note 6.

3) On page 9 they write “The excellent match between the two curves unambiguously confirms that the signal arises from spin-to-charge conversion”. I disagree. In the linear transport regime, where most of these experiments are done, reciprocity relations for four terminal measurements should always be obeyed, no matter what the mechanisms are.

We are a bit confused by this statement, because this sentence refers to Fig. 4b, where we overlapped the ISHE signal with the $\sin\theta$. The overlapping of the two curves is not to prove the reciprocity relation, but an additional evidence for the spin-to-charge conversion origin of the signal. In this figure, we compare the two curves obtained from different measurements: i) Non-local Hanle where injection and detection use the Co electrodes (also shown in Fig. 2d), and ii) Inverse spin Hall effect where injection uses the Co electrode and detection uses Pt (following the configuration in Fig. 4a). The good match between these two curves is a powerful proof of the origin of the observed signal. In fact, this match was used in the first demonstration of electrical detection of the inverse spin Hall effect by Valenzuela and Tinkham [Nature 442, 176 (2006)].

We believe that the referee had in mind Fig. 4c, where the spin Hall effect and inverse spin Hall effect curves are shown and the reciprocity relations are obeyed, but our sentence is not referring to this figure.

We have modified this sentence in the main text to avoid confusion.

So the observation of reciprocity does not exclude other mechanisms, such as stray fields. They can be relevant because when the Co ferromagnet is magnetized in the y-direction, it will produce stray fields in the graphene close to the ends of the electrodes. These will have components in the z-direction, so they could produce magneto resistance effects which would scale with the magnetization in the y-direction of the Co. I did not see a discussion of this. Can the authors please argue why the effects of stray fields are not relevant?

First of all would like to emphasize that the technique we employ and leads to the observation of the reciprocity of ISHE and SHE has been used in many other systems before, being now a well-established method to detect and quantify spin-to-charge current conversion [see for instance Refs. 8,13,15,21,47-51 of the main paper or the recently published work in Nat. Mater. **14**, 675 (2015)]. Our results in Figs. 4b and 4c, thus, are considered a standard demonstration of the ISHE and SHE using the spin absorption technique.

However, we agree that out-of-plane stray fields are probably present close to the end of the Co electrodes when magnetized in the y-direction. Nevertheless, spurious magnetoresistive effects do not have the proper symmetry to reproduce the observed ISHE signals:

i) Magnetoresistance is always an even function of the magnetic field, whereas our signal is an odd function of the field.

ii) The ubiquitous van der Pauw currents spreading in the graphene channel could experience Hall effect (which is an odd function of field) with the out-of-plane stray fields. However, the current basically runs parallel to the Co injector electrode, which would lead to a Hall voltage along the graphene channel. This configuration cannot give a voltage drop along the Pt wire, which is perpendicular to the graphene channel. It is true in some regions the van der Pauw currents can run perpendicular to the Co injector electrode, but you have it in both directions, effectively cancelling out the Hall voltage contribution of these regions.

In any case, one could argue that the symmetry of case ii) can be lost in real experiments with devices with imperfect geometries. Therefore we performed a clear control experiment by replacing Pt with a metal without spin Hall effect, Cu. In this case, no ISHE signal is expected, but any signal coming from stray fields should appear anyway. The results of the measurement are shown above (answer of point 3 by Referee 1). The clear absence of signal in the control experiment ultimately rules out the possibility of stray field contribution. The same negative results are measured in 4 other control samples. These control experiments are yet another strong experimental evidence to prove that spin-to-charge conversion is the origin in our experiment with Pt.

This control experiment is added and discussed in a new Supplementary Note 2.

4) According to the authors the spin-charge- conversion at the Pt strip happens because of spin which are polarized in the y-direction enter the strip in the z-direction, thus producing a charge current in the x-direction. However I would not expect this to happen uniformly over the entire Pt/graphene interface. Because this is a metallic interface I would expect that the spins will already enter at the beginning of the strip. It is not clear if the authors have taken this into account, or whether it changes their conclusions. Can the authors comment on this?

We agree with the referee that this is an important point, which essentially questions how valid is the 1D approximation of the used spin diffusion equations to evaluate the spin absorption in our real devices. This issue has been extensively studied by Laczkowski et al. [Phys. Rev. B, 92, 214405 (2015)]. They report that the width of the absorber material is of paramount importance for the validity of Supplementary Equation 7. The authors observe that, when this width becomes comparable to the spin diffusion length of the material where the spins propagate, Supplementary Equation 7 is not accurate anymore. In our case, the spin diffusion length of graphene is much longer (1200 nm) than the Pt wire width (200 nm) and, thus, the calculated values are not affected.

An additional proof is the spin absorption experiments by Pt using Cu as a spin transport channel by one of us [Niimi et al., PRB **89**, 054401 (2014)]. The spin diffusion length of Cu is similar to that of graphene, and the Pt/Cu interface is a metallic as in our Pt/graphene case. The experiments are analyzed using both the 1D approximation and a complete 3D model. The obtained spin diffusion length and spin Hall angle in 1D have a very small deviation with respect the 3D case, confirming the validity of our approach.

We have added this discussion and the mentioned key references to new Supplementary Note 4.

In summary, I am willing to give a positive recommendation, but I first would like to see a convincing reply to my points 3) and 4)

We believe we have provided a full and convincing explanation to points 3) and 4). We hope the referee will be convinced of the validity of our results and conclusions and, accordingly, will recommend the paper for publication.

REVIEWERS' COMMENTS:

Reviewer #1 (Remarks to the Author):

The authors have carefully answered all referees' questions, especially add control experiments by replacing Pt with Cu contact. No ISHE effect is observed in such control sample, which validates their conclusion. Although the authors claim that due to lack of reproducibility of interfacial barrier, distance dependent measurement is not reliable, I still wish to see these results in their future measurements. In the discussion or introduction, I think it is important to add one sentence in the main text: "The concept of using charge-to-spin to inject spin and spin-to-charge conversion to detect spin in graphene based 2D materials could open future applications of all electrical control of spin manipulation without magnetic field." In summary, I recommend to publish this very interesting result.

Reviewer #2 (Remarks to the Author):

The authors have carefully replied to all remarks of the referees. They have done an extra control experiment, and have made appropriate additions to the manuscript and supplementary information. I can now recommend publication.

RESPONSE TO REVIEWERS' COMMENTS:

Reviewer #1 (Remarks to the Author):

“The authors have carefully answered all referees’ questions, especially add control experiments by replacing Pt with Cu contact. No ISHE effect is observed in such control sample, which validates their conclusion. Although the authors claim that due to lack of reproducibility of interfacial barrier, distance dependent measurement is not reliable, I still wish to see these results in their future measurements. In the discussion or introduction, I think it is important to add one sentence in the main text: “The concept of using charge-to-spin to inject spin and spin-to-charge conversion to detect spin in graphene based 2D materials could open future applications of all electrical control of spin manipulation without magnetic field.” In summary, I recommend to publish this very interesting result.”

We thank the referee for recognising the significant implication of our work and we have added the above content appropriately in the manuscript.

The end of the abstract now reads:

“Our approach opens up exciting opportunities towards the implementation of spin-orbit-based logic circuits and all electrical control of spin information without magnetic field.”

The end of the introduction now reads:

“Our concept of using charge-to-spin conversion to inject spin currents and spin-to-charge conversion to detect spin currents in graphene-based devices could open future applications of all electrical control of spin information without magnetic field.”

Reviewer #2 (Remarks to the Author):

“The authors have carefully replied to all remarks of the referees. They have done an extra control experiment, and have made appropriate additions to the manuscript and supplementary information. I can now recommend publication.”

We thank the referee for the positive feedback.